# VARIATIONAL SGD: DROPOUT, GENERALIZATION AND CRITICAL POINT AT THE END OF CONVEXITY

## ABSTRACT

The goal of the paper is to propose an algorithm for learning the most generalizable solution from given training data. It is shown that Bayesian approach leads to a solution that dependent on statistics of training data and not on particular samples. The solution is stable under perturbations of training data because it is defined by an integral contribution of multiple maxima of the likelihood and not by a single global maximum. Specifically, the Bayesian probability distribution of parameters (weights) of a probabilistic model given by a neural network is estimated via recurrent variational approximations. Derived recurrent update rules correspond to SGD-type rules for finding a minimum of an effective loss that is an average of an original negative log-likelihood over the Gaussian distributions of weights, which makes it a function of means and variances. The effective loss is convex for large variances and non-convex in the limit of small variances. Among stationary solutions of the update rules there are trivial solutions with zero variances at local minima of the original loss and a single non-trivial solution with finite variances that is a critical point at the end of convexity of the effective loss in the mean-variance space. At the critical point both first- and second-order gradients of the effective loss w.r.t. means are zero. The empirical study confirms that the critical point represents the most generalizable solution. While the location of the critical point in the weight space depends on specifics of the used probabilistic model some properties at the critical point are universal and model independent.

## 1 INTRODUCTION

Finding a generalizable solution is a critical problem for any machine learning task. The ultimate goal of learning from the available ground truths is to make a good prediction for new data. The Bayesian method is a very powerful approach that gives a probabilistic measure of the ability of a proposed model to predict by estimating how well the model predicts known data.

The accuracy of the predictions depends on how the found solution is able to overcome a sampling bias to avoid overfitting for given particular samples of training data.

Specifically, in Bayesian method predictions of labels $y$ for an input $x$ are made by using a probabilistic model, for certainty a neural network, which defines a function parametrized by weights $w$ that allows computing probabilities $P(y|x, w)$ for each weight point. Each weight point contributes to predicted probabilities of labels $Prob(y|x)$ in accordance with probability distribution of weights. The distribution of weights is learned from a known training data set $\{x_n, y_n; n = 1..N\}$ and its prior probability distribution $P_0(w)$ in the following way:

$$Prob(y|x) = \int_w P(y|x, w) P_0(w) \prod_{n=1}^{N} P(y_n|x_n, w) / \int_w P_0(w) \prod_{n=1}^{N} P(y_n|x_n, w) \tag{1}$$

Here the predicted probability $Prob(y|x)$ is an average of the model probability $P(y|x, w)$ at a weight $w$ over the learned weight distribution. To make predictions we are only interested in a method that allows to find averages in eq.(1) and not absolute values of the integrals. According to mean value theorem (Cauchy (1813), also in *Encyclopedia of Mathematics* Hazewinkel (1994)) values of the averages can be represented by a single point, which in our case means that there

is a single point in the weight space $w_0$ that represents a result of computing the integrals, so $Prob(y|x) = P(y|x, w_0)$. That point $w_0$ is a solution of the training of the neural network.

A standard approach to get the solution is a maximum likelihood method that finds a maximum of the integrand. However, there are some cases when the maximum likelihood fails to represent main contribution to the integral by weights. Consider this example: if log-likelihood for $N$ data samples has a maximum at some weight point $w_1$, then in general its first derivative by weights is zero, second derivative is negative and proportional to $N$, so corresponding Gaussian integral by the weights is proportional to $N^{-d/2}$, where $d$ is number of weights. This will change if there is a flat maximum, which has not only first but also second and third derivatives equal to zero. In this case the integral is proportional to $N^{-d/4}$. For large number of samples the flat maximum makes the most significant contribution to the integral by weights:

$$I_1 = P_1^N \int_w e^{-NC_2 w^2} \propto P_1^N \times N^{-d/2}, I_2 = P_2^N \int_w e^{-NC_4 w^4} \propto P_2^N \times N^{-d/4},$$

and $I_2/I_1 \propto (P_2/P_1)^N \times N^{d/4}$. For a typical case when the number of weights $d \sim N$ and average sample probabilities at maxima are comparable $O(P_1) \sim O(P_2)$ the integral around flat maximum $I_2$ is always bigger than the integral around narrow maximum $I_1$, unless $P_2$ is zero.

While in general a likelihood has a number of regular local maxima and no flat maximum the effect of integration over multiple frequent local maxima can result in an effective flat maximum that defines a solution.

We argue that any local or global maximum of likelihood gives a wrong solution that is not generalizable and so makes inaccurate predictions, because the locations for the global maximum and local maxima depend on specific samples in the training data and any modification of it by adding or removing samples will change the solution (Zhang et al., 2017). Instead we will show that there is another solution that more associated with properties of the distribution of training data and less with particular samples.

The purpose of this paper is to show that the effective flat maximum always exists for specific parameters of prior weight distribution $P_0(w)$ (regularization parameters) and corresponding solution is the most generalizable solution that can be found in training. We show that the solution is a critical point in an effective loss that represents the result of integration over the weights.

In the next sections we derive the algorithm for the optimizer for finding the critical point solution and analyze properties of the solutions. The empirical study is outside of the scope of the paper and will be presented separately.

For simplicity we use same notations for a vector of weights and its components, as well as corresponding parameters of distributions of weights because all weight components are independent in our consideration and it is clear from context when it is a vector or its component.

## 2 METHOD FOR COMPUTING THE INTEGRALS

We use a recurrent approach for estimating the integrals. First, we represent a probability of each training sample as a product of factors close to one, $P(y|x, w) = (1 + 1/T \ln P(y|x, w))^T$, where free parameter $T \gg 1$ is a number of epochs:

$$\int_w P_0(w) \prod_{n=1}^N P(y_n|x_n, w) = \int_w P_0(w) \prod_{n=1}^N \left(1 + \frac{1}{T} \ln P(y_n|x_n, w)\right)^T,$$

then model each factor as a product of Gaussian distributions $Q(w|\mu, \sigma)$ one for each component of weight vector.

For each iteration a running prior distribution of weights is updated by absorbing a single factor to produce a new prior

$$Q_t(w) \left(1 + \frac{1}{T} \ln P(y|x, w)\right) \to Q_{t+1}(w), \text{ with } Q_0(w) = P_0(w) \text{ and } Q(w|\mu, \sigma) = \frac{e^{-\frac{(w-\mu)^2}{2\sigma^2}}}{\sqrt{2\pi\sigma^2}}.$$

Specifically, we do the following:

First, let's enumerate all factors for all data samples

$$\prod_{n=1}^{N} \left(1 + \frac{1}{T} \ln P(y_n|x_n, w)\right)^T = \prod_{t=1}^{N \times T} \left(1 + \frac{1}{T} \ln P(y_t|x_t, w)\right)$$

Then, under the integral by weights we use an identical re-writing for a product of a prior $Q_t(w)$ and one of the factors and a new prior $Q_{t+1}(w)$ and normalization factor $N_t$

$$\int_w Q_t(w) \left(1 + \frac{1}{T} \ln P(y_t|x_t, w)\right) (\ldots) = N_t \int_w \left[\frac{R_t(w)}{Q_{t+1}(w)}\right] Q_{t+1}(w) (\ldots),$$

where normalization factor

$$N_t = \int_w Q_t(w) \left(1 + \frac{1}{T} \ln P(y_t|x_t, w)\right)$$

and distribution

$$R_t(w) = \frac{Q_t(w) \left(1 + \frac{1}{T} \ln P(y_t|x_t, w)\right)}{N_t}.$$

Finally, we make an approximation by replacing ratio of distributions $R_t(w)/Q_{t+1}(w)$ by its mean for distribution $Q_{t+1}$ which is equal to 1. Then the iterations are repeated until all factors from probabilities of data are replaced by a single final Gaussian distribution and some normalization factor.

To minimize the introduced error on each iteration we select the means and variances of the new prior $Q_{t+1}$ to minimize variance of the ratio $R_t/Q_{t+1}$. The variance is equal to

$$\text{Var}_{Q_{t+1}}[R_t/Q_{t+1}] = \int_w \left[\frac{R_t(w)}{Q_{t+1}(w)}\right]^2 Q_{t+1}(w) - 1 = \mathbb{E}_{R_t}\left[\frac{R_t(w)}{Q_{t+1}(w)}\right] - 1.$$

The lower bound of the variance is expressible via KL divergence (Kullback & Leibler (1951)) of $R_t$ and $Q_{t+1}$

$$\text{Var}_{Q_{t+1}}[R_t/Q_{t+1}] \geq \exp\left(\int_w R_t(w) \log \frac{R_t(w)}{Q_{t+1}(w)}\right) - 1 = \exp\left(D_{KL}(R_t \| Q_{t+1})\right) - 1.$$

Finding the minimum of KL divergence is equivalent to minimizing the lower bound of the variance of the ratio of $R_t/Q_{t+1}$ which leads to equations

$$\frac{\partial}{\partial \mu_{t+1}}, \frac{\partial}{\partial \sigma_{t+1}} \int_w Q(w|\mu_t, \sigma_t) \left(1 + \frac{1}{T} \ln P(y|x, w)\right) \ln Q(w|\mu_{t+1}, \sigma_{t+1}) = 0.$$

Then solving the above equations gives the update rules for means and variances of each weight component

$$\mu_{t+1} = \mu_t + \frac{\sigma_t^2}{NT} \frac{\partial}{\partial \mu_t} \left\langle \ln P(w) \right\rangle_{\mu_t, \sigma_t}, \quad \frac{1}{\sigma_{t+1}^2} = \frac{1}{\sigma_t^2} - \frac{1}{NT} \frac{\partial^2}{\partial \mu_t^2} \left\langle \ln P(w) \right\rangle_{\mu_t, \sigma_t}, \quad (2)$$

where averages are defined as $\left\langle A(w) \right\rangle_{\mu, \sigma} = \int_w Q(w|\mu, \sigma) A(w)$ and averages of gradients by weights are equal to gradients of averages by means $\left\langle \frac{\partial}{\partial w} A(w) \right\rangle_{\mu, \sigma} = \frac{\partial}{\partial \mu} \left\langle A(w) \right\rangle_{\mu, \sigma}$.

Another useful identity allows to replace second gradient of $(\mu, \sigma)$-average w.r.t mean on first gradient by variance

$$\frac{\partial}{\partial \sigma^2} \left\langle A(w) \right\rangle_{\mu, \sigma} = \frac{1}{2} \frac{\partial^2}{\partial \mu^2} \left\langle A(w) \right\rangle_{\mu, \sigma}. \quad (3)$$

For a full batch the log of probability of data in eqs.(2) $\ln P(w) = \sum_n^N \ln P(y_n|x_n, w)$, while for a minibatch the sum goes over the size of the minibatch.

In eqs.(2) we used rescaled variances $\sigma^2 \to \sigma^2/N$ so all gradients are normalized per data sample. With the rescaled variances the prior distribution of weights is a product over all weight dimensions $d$

$$P_0(w|\mu_0, \sigma_0) = N^{d/2} \prod_{i=1}^{d} \left( \frac{e^{-N\frac{(w_i - \mu_{i,0})^2}{2\sigma_{i,0}^2}}}{\sqrt{2\pi\sigma_{i,0}^2}} \right).$$

The index $t$ enumerates iterations over all minibatches in all epochs. Then for a minibatch size one the total number of iterations equals to a number of epochs $T$ times number of data samples $N$ in a training set.

By recursively applying the update rules in eqs.(2) over $N$ samples and $T$ epochs we obtain the approximation of Bayesian integrals that allows to compute averages

$$\int_w P_0(w) \prod_n^N P(y_n|x_n, w)(\dots) \approx \int_w Q_{t_{max}}(w)(\dots) \times \exp\left( \frac{1}{T} \sum_t^{t_{max}} \int_w Q_t(w) \ln P_t(w) \right).$$

It is important to emphasize that the averages are defined by distribution $Q_{t_{max}}(w)$ after a finite number of iterations $t_{max}$, which for the minibatch one is equal to a product $N \times T$ and not by distribution at the infinite number of iterations $Q_\infty(w)$. Number of epochs $T$ controls the accuracy of the factorized representation and in practical computing any $T$ larger than 10 or 100 is good enough.

The prediction probability in eq.(1) is defined by weight $w_0$ that is a mean $\mu_{t_{max}}$ of distribution $Q_{t_{max}(w)}$

$$Prob(y|x) \approx P(y|x, \mu_{t_{max}}).$$

That mean $\mu_{t_{max}}$ is a final point of iterations in eqs.(2). The trajectory in mean-variance space is defined by starting point $(\mu_0, \sigma_0)$, the mean and variance of a prior distribution of weights in eq.(1) which are regularization parameters: $P_0(w) = Q(w|\mu_0, \sigma_0)$.

## 3 RESULTS

Before going into detailed analysis of eqs.(2) let's formulate the results in the form of the following statements.

The update rules above are solving SGD-type optimization problem for an effective loss given by Gaussian average $L(\mu, \sigma) = -\int_w Q(w|\mu, \sigma) \sum_n \ln P(y_n|x_n, w)$ for each mean-variance point $(\mu, \sigma)$.

The following statements have been proved:

1. The effective loss $L(\mu, \sigma)$ is convex for large variances $\sigma^2$. In particular, it is true for any neural network with ReLU activations and L1, L2 or cross entropy loss functions. For a convex effective loss its second-order gradient by mean is positive.

2. The effective loss is converging to the original loss in the limit of small variances where it is generally non-convex (excluding completely trivial linear cases).

3. For each mean there is a critical variance $\sigma_c^2$ that separates convex effective loss from non-convex. At the critical variance second-order gradient of effective loss by mean is zero.

4. There are trivial stationary solutions of the update rules that correspond to zero variances and zero gradients of loss w.r.t. weights. These solutions are unstable when training set is modified because they correspond to narrow minima that are changing drastically as data change (Zhang et al., 2017). Because second-order gradients by weights are large and positive, changes in loss are second-order by weights and changes in solutions are at least linear and often result in jumps to a new location.

5. There is a non-trivial stationary solution at critical point at the end of convexity where both first-order and second-order gradients of effective loss are zero. That solution is much less sensitive to changes in data. It is responding by cubic changes in loss and quadratic changes in solutions and due to convexity there is no jumps. For that reason the critical point solution is the most generalizable solution as well as most stable against adversarial perturbations (Goodfellow et al., 2014).

6. Trajectories in mean-variance space that follow from update rules show universal behavior in vicinity of the critical point. Analysis shows that not all trajectories are converging to the critical point. Typical trajectory that starts in convex area moves toward the critical point until it may cross to non-convex area then it moves away from critical point and finally ends as a trivial solution in a local minimum.

7. Approximating Gaussian averaging by sampling in weight space results in dropout method also known as "fast dropout" (Wang & Manning, 2013) and "dropconnect" (Wan et al., 2013). The update rules completely define the dropout rate for each weight component. With averaging via dropout the critical point and the convexity of effective loss exist only in an unreachable limit, nevertheless it allows to find a solution that is close to the critical point.

# 4  ANALYSIS OF A SIMPLE TWO-MINIMUM MODEL

To understand better an effect of averaging let's consider a simple polynomial case where loss $l(w)$ as a function of weights has two symmetrical minima at points $(-a, +a)$:

$$l(w) = (w^2 - a^2)^2.$$

Then the effective loss is Gaussian average of the loss above $L(\mu, \sigma) = \langle l(w) \rangle_{\mu,\sigma}$, specifically

$$L(\mu, \sigma) = \int_w \frac{e^{-\frac{(w-\mu)^2}{2\sigma^2}}}{\sqrt{2\pi\sigma^2}} (w^2 - a^2)^2 = (\mu^2 - a^2)^2 + 12\sigma^2 \left( \frac{\mu^2}{2} + \frac{\sigma^2}{4} - \frac{a^2}{6} \right). \qquad (4)$$

And the first and second gradients of the effective loss are

$$\frac{\partial L}{\partial \mu} = 4\mu \left( \mu^2 - a^2 + 3\sigma^2 \right), \quad \frac{\partial^2 L}{\partial \mu^2} = 12\mu^2 + 4 \left( 3\sigma^2 - a^2 \right). \qquad (5)$$

When variance is large, $\sigma^2 > a^2/3$, second gradient of the effective loss $L$ w.r.t. $\mu$ is positive and there is only one minimum at $\mu = 0$. When variance is small $\sigma^2 < a^2/3$, the effective loss has a maximum at $\mu = 0$ and two minima at $\mu = \pm\sqrt{a^2 - 3\sigma^2}$.

When $\sigma^2 = a^2/3$ there is a critical point at $\mu = 0$ where both first and second gradients of the effective loss w.r.t. $\mu$ are zero: $\frac{\partial L}{\partial \mu} = 0, \frac{\partial^2 L}{\partial \mu^2} = 0$.

By using the update rules from eqs.(2) where average log of probability is equal to the negative effective loss $L = -\langle \ln P \rangle$ with first and second gradients defined above in eqs.(5) we can consider trajectories in the mean-variance space:

$$\mu_{t+1} = \mu_t - \frac{\sigma_t^2}{T} \frac{\partial L(\mu_t, \sigma_t)}{\partial \mu_t}, \quad \frac{1}{\sigma_{t+1}^2} = \frac{1}{\sigma_t^2} + \frac{1}{T} \frac{\partial^2 L(\mu_t, \sigma_t)}{\partial \mu_t^2}$$

We can see that the critical point is a saddle point in the mean-variance space. Any trajectory that is missing a critical point after an infinite number of iterations ends in a local minimum of an effective loss with zero variance $\sigma^2$, which is an original local minimum.

There are only two trajectories starting with $\mu = 0$ from convex area $(3\sigma^2 > a^2)$ and non-convex area $(3\sigma^2 < a^2)$ that after a large enough number of iterations will always converge to the critical point.

However, for a finite number of iterations there is an area of starting points $(\mu_0, \sigma_0)$ that defines trajectories with end points arbitrary close to a critical point.

## 5    ANALYSIS OF A GENERAL CASE

Let's consider a multilayered neural network where predictions $y$ are computed via ReLU activations and loss function $l(y)$ is a convex function of predictions. For each weight $w$ predictions $y$ are piecewise functions of the weight. Then second gradient of loss function by weight is

$$\frac{\partial^2 l(y(w))}{\partial w^2} = \frac{\partial^2 l(y)}{\partial y^2} \left(\frac{\partial y(w)}{\partial w}\right)^2 + \frac{\partial l(y)}{\partial y} \frac{\partial^2 y(w)}{\partial w^2}.$$

There are two terms in second gradient of loss by weight: first is always positive due to convexity of $l(y)$ and second contains second gradient of predictions. Average of the first term is always positive. Average of the second term could be positive or negative.

Due to piecewise weight dependency second gradient of predictions $\partial^2 y / \partial w^2$ is singular and not zero, however for any network with finite number of layers predictions $y$ are linear functions of the weight when the value of the weight goes to infinity. For that reason second gradient of predictions for large values of the weight is zero and second gradient of loss by weight is positive for large weight values.

If variance of Gaussian distribution of the weight is very large then averages are defined by contributions of large weight values and then average of the second gradient of loss by weight is positive.

$$\text{if} \quad \sigma \to \infty, \quad \langle\frac{\partial^2 l(w)}{\partial w^2}\rangle_{\mu,\sigma} > 0 \quad \text{and} \quad \frac{\partial^2 L(\mu,\sigma)}{\partial \mu^2} > 0.$$

That proves the convexity of the effective loss for large variances.

On other side when the variance goes to zero the effective loss converges to original loss which generally non-convex. Because an effective loss is a continuous function of the variance there exists a critical value of the variance where second gradient of the effective loss may have zero at some mean. That means the mean-variance plane could be divided on two convex and non-convex areas.

These areas are separated by the line on mean-variance plane where second gradient of the effective loss w.r.t. means is zero.

The general properties of the critical point are universal due to the identity in eq.(3). Let's consider an expansion of the effective loss $L(\mu,\sigma)$ in a vicinity of a point $\mu_c, \sigma_c$ where both first gradients of $L$ by mean and variance are zero:

$$L \approx a_2(\mu - \mu_c)^2 + a_3(\mu - \mu_c)^3 + b_2(\sigma^2 - \sigma_c^2)^2 + c(\mu - \mu_c)(\sigma^2 - \sigma_c^2) + \dots,$$

so

$$\frac{\partial L}{\partial \mu} = 2a_2(\mu - \mu_c) + 3a_3(\mu - \mu_c)^2 + c(\sigma^2 - \sigma_c^2), \quad \frac{\partial L}{\partial \sigma^2} = 2b_2(\sigma^2 - \sigma_c^2) + c(\mu - \mu_c)$$

Then using the identity in equation (3) we can see that

$$\frac{1}{2}\frac{\partial^2 L}{\partial \mu^2} = a_2 + 3a_3(\mu - \mu_c) \quad \text{and} \quad a_2 = 0$$

because zero-order term by $(\mu - \mu_c)$ in $\frac{\partial L}{\partial \sigma^2}$ is zero. Also it gives $c = 3a_3$.

Due to the identity (3) the universal structure of the effective loss at the critical point is as follows

$$L \approx a_3(\mu - \mu_c)^3 + b_2(\sigma^2 - \sigma_c^2)^2 + 3a_3(\mu - \mu_c)(\sigma^2 - \sigma_c^2) + \dots$$

Essentially, the condition (3) makes a regular minimum of the effective loss to be a critical point.

At critical point both first and second gradients by means of the effective loss are zero. At critical point the effective loss is a flat function of the means. That makes the solution stable when training data set is modified unlike any maximum log-likelihood solutions that changes when data changes. For that reason the critical point solution is most generalizable.

# 6 GENERALIZATION ERROR FOR A SIMPLE TWO-MINIMUM MODEL

Any available data set is a collection of samples from an unknown true distribution. The empirical loss is defined as loss per sample for sampled data for a given weight point. The expected loss is average over unknown true distribution of data at a given weight point.

In a two-minimum model loss for a data point is $l(w) = (w^2 - a^2)^2$, where $a$ is a sample specific parameter. The empirical loss per sample for $m$ samples is

$$L_{emp}(w) = \frac{1}{m} \sum_{n=1}^{m} (w^2 - a_n^2)^2 = w^4 - 2w^2 \frac{1}{m} \sum_{n=1}^{m} a_n^2 + \frac{1}{m} \sum_{n=1}^{m} a_n^4,$$

while the expected loss per data point for the same weight point is

$$L_{exp}(w) = \mathbb{E}\left[(w^2 - a^2)^2\right] = w^4 - 2w^2 \mathbb{E}\left[a^2\right] + \mathbb{E}\left[a^4\right].$$

The generalization error is a difference between an expected loss and empirical loss for the same solution $w$:

$$GErr(w) = L_{exp}(w) - L_{emp}(w).$$

For a given weight point the expectation of the generalization error is zero: $\mathbb{E}\left[GErr(w)\right] = \mathbb{E}\left[L_{exp}(w)\right] - \left[L_{emp}(w)\right] = 0$.

However, we only can find solution for a minimum of the empirical loss.

$$\frac{\partial L_{emp}(w)}{\partial w} = 0, w_{emp}^2 = \frac{1}{m} \sum_{n=1}^{m} a_n^2.$$

Due to sampling a minimum loss solution for the empirical loss depends on specific samples. For that reason the expectation of the generalization error for a minimum of the empirical loss solution is not zero and is equal to

$$\mathbb{E}\left[GErr(w_{emp})\right] = 2\mathbb{E}\left[\frac{1}{m} \sum_{n=1}^{m} a_n^2 \left(\frac{1}{m} \sum_{n=1}^{m} a_n^2 - \mathbb{E}\left[a^2\right]\right)\right] = \frac{2}{m} \mathbb{E}\left[(a^2 - \mathbb{E}\left[a^2\right])^2\right]. \quad (6)$$

The expectation of the generalization error is positive which reflects overfitting and goes to zero when number of samples $m$ goes to infinity.

Now, let's compute generalization error for the effective loss (4) for $m$ data samples. The solution for empirical effective loss are found from equations

$$\frac{\partial L_{emp}(\mu, \sigma)}{\partial \mu} = 0, \frac{\partial L_{emp}(\mu, \sigma)}{\partial \sigma} = 0.$$

There are trivial solutions with zero variance $\sigma_0 = 0, \mu_0^2 = \frac{1}{m} \sum_{n=1}^{m} a_n^2$. These solutions have the same expected generalization error as in eq.(6): $\mathbb{E}\left[GErr_{eff}(\mu_0, \sigma_0)\right] = 2\mathbb{E}\left[(a^2 - \mathbb{E}\left[a^2\right])^2\right]/m$.

And there is a critical point solution with non-zero variance: $\mu_c = 0, 3\sigma_c^2 = \frac{1}{m} \sum_{n=1}^{m} a_n^2$. That solution gives the expected generalization error that is 3 times smaller than the expected generalization error for trivial solutions:

$$\mathbb{E}\left[GErr_{eff}(\mu_c, \sigma_c)\right] = \frac{2}{m}\mathbb{E}\left[(a^2 - \mathbb{E}\left[a^2\right])^2\right]/3 = \mathbb{E}\left[GErr_{eff}(\mu_0, \sigma_0)\right]/3.$$

This example supports the claim of the paper that the critical point solution is more generalizable than a trivial minimum loss solution.

## 7  CONCLUSIONS

In the paper we consider a learning of a predictive model from training data by approximately computing Bayesian integral over weights - the parameters of the model.

By using recurrent variational approximations with Gaussian weight distributions we are able to find a solution - a single point in weight space that represents an effect of averaging over distribution of weights in the Bayesian integrals.

We show that this approach leads to SGD-type optimization problem for an effective loss in mean-variance space. For each mean-variance point the effective loss is defined by average of the log-likelihood over Gaussian distribution at same mean-variance point. Due to averaging the effective loss and its gradients of any order are continuous function of means even for ReLU based neural networks.

The recurrent update rules define trajectories in mean-variance space. Starting points of the trajectories are defined by regularization parameters, which are parameters of the Gaussian weight prior in Bayesian integrals.

It is shown that there are two types of stationary solutions of the update rules. First solution type corresponds to local minima of the original loss or maxima of the log-likelihood. Second solution type is a critical point in mean-variance space that is a result of the integration over multiple maxima of the log-likelihood.

At the critical point both first and second gradient of the effective loss are zero. That leads to stability of the solution against perturbations of the training data set due to addition or removal data samples or via creation of adversarial examples.

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
