# OpenReview forum: "VARIATIONAL SGD: DROPOUT , GENERALIZATION AND CRITICAL POINT AT THE END OF CONVEXITY"
_ICLR.cc/2019/Conference_

### Official Review · AnonReviewer1 · 2018-10-29
**unclear, critically ill-written paper**

**Rating:** 2
**Confidence:** 5

**Review:**

Presentation of the work is critically weak and I failed to understand the objective and contributions of the paper (despite a solid knowledge in Bayesian inference). They are many editing problems and the English is problematic, but most importantly the writing fails to properly introduce the problem, the objective and solutions.

---

### Official Review · AnonReviewer3 · 2018-11-02
**VARIATIONAL SGD**

**Rating:** 2
**Confidence:** 4

**Review:**

Summary of the paper:
The paper proposes an algorithm to find solution to the maximum likelihood problem that could generalize well. The paper argues from the point of view that purely optimizing over the likelihood could result in solution that corresponds to poor local minimum which does not generalize well. By introducing a certain prior on weight, there exists a solution that could generalize. The solution arrived by introducing the prior makes it stable under perturbations of the training data. Recurrent update rules are derived for computing the integrals and hence the solution could be calculated. The authors discuss about the convexity of the effective loss when the variance is large.

The paper itself is very bad in its presentation. In terms of technical presentation, it is missing a lot of details, which makes reading and understanding the paper very hard.
1.	It does not come with any proper literature review and introduction to the formulation of the problem.
2.	The presentation of the methodology is also missing a lot of the explanation for many of the details used in the method. For example, in section 2, I do not quite understand the reasoning behind setting the probability P(y|x,w) = (1+1/T lnP(y|x,w))^T. Also, why R_t(w)/Q_(t+1)(w) could be approximated by 1.
3.	The theoretical results come in plain words without proper mathematical presentation and the proofs for the statements are not well organized. The correspondence between the proofs and statements are not clear.
4.	There seems to be no experiments conducted to support the practical use of the method proposed in the paper.

Overall, I feel the paper is not ready for publication as a conference paper. The lack of details especially for the technical presentation part make it very hard to read. And the presentation of the results seem to be short of clarity and organization. Further, no experiments showing the practicality of the method are included in the paper.

---

> ### Author Response · Authors · 2018-11-26
> **Please see answers to some of your questions in notes above.**
>
> 1. The reviewer is right about missing proper literature review.
> 2. Please see above the answer to that question.
> 3. Hope the provided clarifications in notes above are adding proper mathematical presentations.
> 4. The conducted experiments are not fitting the capacity of the paper and will be published separately.

---

### Official Review · AnonReviewer2 · 2018-11-04
**Study of the landscape of the effective loss**

**Rating:** 4
**Confidence:** 3

**Review:**

Summary: Non-convex learning problems can have multiple solutions with different generalization properties, thus it is important to find solutions that generalize well. The goal of this paper is to derive an algorithm for finding a solution to the learning problem with the best possible generalization properties. This is achieved by using a Bayesian approach in which the parameters w (e.g., weights of a network) are random variables and the effective loss (integral wrt to w) is minimized in lieu of the usual loss. The paper assumes that each component of the weight vector w is Gaussian and derives a formula for updating the mean and covariance of said Gaussians (this is an SGD method). The paper claims that the resulting effective loss is convex for large variances (sigma > threshold), nonconvex for small variances (sigma < threshold), and converges to original loss as sigma goes to zero. The paper also claims that when sigma=0 there are trivial solutions that are unstable as data changes, but that when sigma=threshold (assuming this is what is meant by end of convexity) there are non-trivial solutions that are less sensitive to data changes and hence the most generalizable.

Comments: the goal of the paper (finding minima that generalize well) is an excellent one. But the paper is not clearly written and appears to oversell the contribution. In particular, the title speaks about SGD, dropout, generalization and critical points “at the end of convexity”. Naturally, a reader is inclined to think that the paper will study SGD and dropout for deep learning and analyze generalization properties of the solutions found by those methods. In reality, there is very little in the paper about SGD, dropout, and generalization. The connection with SGD is merely because the method for updating mu and sigma is an SGD method. The connection with dropout is mentioned in passing in one paragraph and it is not very clear. The connection with generalization is claimed but never quite explained (there are no generalization bounds in the paper). As far as understand, the paper considers the minimization of the effective loss, uses a Gaussian approximation for computing the effective loss, and focuses primarily on the characterization of convexity as a function of sigma as well as a characterization of the critical points depending on whether the effective loss is convex (sigma above a threshold) or not (sigma below the threshold). The main claim appears to be that critical points at the critical threshold lead to solutions that generalize well, but a detailed explanation of why this is the case isn't given. If my digest of the paper is the correct one, then modifying the title, abstract and intro to make this clear would have helped a lot.

Beyond the high-level lack of clarity about the contribution of the paper, the writing lacks precision and rigor, and many things are undefined (though one can figure them out after reading many times back and forth). Specifically:

1) It is not explained why the probability of each training sample can be expressed as a product of factors close to 1, with the product taken over the epochs.

2) It is not explained why each factor can be modeled as a product of Gaussians

3) At nearly the top of page 3, a product over n is substituted by a product over t, with x_n replaced by x_t and so forth, but the total number of products goes up from N to TxN. What is the value of y_{NT} and x_{NxT}? Do the authors mean that y_n should have been replaced by y_{n,t} and we now have two indices? Or do the authors mean that the same mini batch of N samples is reused, and so indices should be corrected accordingly?

4) It is not clear why replacing R_t/Q_{t+1} by 1 is an adequate approximation.

5) At the top of equation 4, there is a product, but no index wrt which the product is taken. Right after it says the index is t, but there is no t in the expression. Should mu_0 be mu_t and similarly for sigma?

In short, a promising direction, but the contribution of the paper appears to be over claimed and the writing of the paper needs significant improvement before the paper can be accepted for publication.

---

> ### Author Response · Authors · 2018-11-26
> **Answers to questions 1-5. Please see a note regarding generalization error above.**
>
> Answers to the questions 1 and 4. It indeed requires more detailed explanation.
>
> The approach used in paper for finding a computable variational approximation based on the following consideration.
>
> Let’s x to be random variable with probability distribution p(x). We want to compute averages for some A(x): <A>_p.
> It is given by integral <A>_p=int_x{p(x)A(x)}, which is in general intractable. Instead we can use another more manageable probability distribution q(x) introduced via identity:
> <A>_p=int_x{q(x)A(x)p(x)/q(x)}=<(p/q)A>_q.
>
> We can get approximation of that average by replacing ratio (p(x)/q(x)) by its mean for distribution q if variance of p/q is small enough:
>
> p/q -> mean_q(p/q)=<p/q>_q = 1;
> var_q(p/q)=<(p/q)^2-1>_q=<p/q>_p - 1.
>
> The mean of p/q is equal to one. To make a good approximation we need to minimize variance of (p/q) by using free parameters of distribution q(x|params). The problem is that <p/q>_p is still intractable. One approach is to use and minimize lower bound of <p/q>_p:
> <p/q>_p >= exp(<log(p/q)>_p) > 1. Here <log(p/q)>_p is a positive KL divergence that we must minimize by selecting appropriate parameters of the distribution q. To do that we need to compute the integral <log(q)>_p=int_x{p(x) log(q(x))}, which for general p(x) is not possible.
>
> We will solve this problem by replacing p with its approximation sequentially one almost identical  factor at the time.
>
> Remember, that our original problem is to compute integral over weights: int_w{p0(w)p1(w)}, where p0 is prior and p1 is a product of model probabilities over all data samples.
>
> That integral is an average of p1 with probability distribution p0: <p1>_p0.
>
> We will sequentially update prior:
>
> First, we factorize p1: p1=(p1^(1/T))^T={exp(log(p1)/T)}^T=(1+log(p1)/T)^T, where T >>1 is large.
> By normalizing product of prior p0 and a single factor (1+log(p1)/T) we make a probability r0=p0(1+log(p1)/T)/<1+log(p1)/T>_p0 and then approximate r0 with q:
>
> <p1>_p0=<{(1+log(p1)/T)^T}(p0/q)>_q=<{(1+log(p1)/T)^(T-1)}{r0/q}>_q *<(1+log(p1)/T)>_p0 =
> <{(1+log(p1)/T)^(T-1)}>_q * <(1+log(p1)/T)>_p0.
>
> Here we replaced r0/q with its mean <r0/q>_q=1, which is one. To make it working we need to minimize var(r0/q)=<(r0/q)^2>_q-1=<r0/q>_r0-1, by selecting parameters of q. Minimizing lower bound of the var(r0/q) is equivalent to minimizing KL divergence
> <log(r0/q)>_r0 which in turn is equivalent to maximizing <(1+log(p1)/T)*log(q)>_p0.
>
> Finding q allows to represent initial integral <p1>_p0=<(1+log(p1)/T)^T>_p0 as <(1+log(p1)/T)^(T-1)>_q, that looks like an original one only with number of factors reduced by one and prior p0 is replaced by prior q.
>
> By repeating this we have recurrent updates: q_t -> q_(t+1), where q_0=p0 and parameters of each new q_t are found by maximizing <(1+log(p1)/T)*log(q_(t+1))>_q_t by parameters of q_(t+1).
>
> The updates repeat exactly T times to approximate initial integral <p1>_p0. We call one such update an epoch because p1 is a product over probabilities of data samples p1=prod_n{p1_n} and log(p1)=sum_n{log(p1_n)} and one such update is a full batch update. If we will do one update per data sample then to complete computing the approximation to the integral it will take N times T updates. Here N is a number of samples and T is a number of epochs.
>
> ============
>
> 2. Why Gaussian?
>
> We can use any probability distribution for variational inference that allows to compute. For the purpose of the paper we can use any distribution with efficient sampling and which has enough parameters to approximate a true distribution.
>
> Freedom to select an approximate distribution is the same freedom we use to select a model for learning statistics of data. We use Gaussian for its special properties and usability for continuous weights. For binary networks we will use another distribution.
>
> We use approximation where each weight is independent, so a whole weight distribution is a product over all weight component distributions. This approach allows to compute efficiently.
>
> If we would like to go beyond a variational approximation we could use MCMC with the approximation developed in the paper as a proposal distribution.
>
> ===============
>
> 3. Indexing
>
> n=1..N is index for data samples. t=1..NxT is index for total NxT iterations. Index t goes over all N data samples T times. In one iteration contribution of only one of T fractions of one of N samples is used. Hope this is clear now.
>
> ==============
>
> 4. See above in 1.
>
> =============
>
> 5. This formula for P0 is the prior of weight and product is taken over each weight component. That product is not related to a following paragraph.

---

### Author Response · Authors · 2018-11-25
**Thank you for your input. Your questions will be answered below in a few comments.**

This note is in support of the claim of the paper regarding generalization for critical point solution.

While general proof is still in work below we consider toy one-dimensional examples that explains how generalization error may be lower at critical point: we show that absolute value of  expectation of generalization error for critical point solution is lower than for “trivial” solutions.

To warm up let’s consider a linear case: loss per data sample L=(w-a)^2, where w is weight and a is sample specific value.
Then loss per sample for m IID samples from data set M is L_m=sum_n{(w-a_n)^2}/m=w^2-2w(sum_n{a}/m)+sum_n{a_n^2}/m.
We will use square brackets for average over an unknown sample distribution: E(z)=[z].
Then expectation of the loss at a given weight w is
E(L)(w)=w^2-2w[a]+[a^2].
Generalization error for a given weight is a difference between loss for data set M and expected loss: gerr(w)=L_m-E(L)=-2w(sum_n{a}/m-[a])+sum_n{a^2}/m-[a^2].
Note that for a given weight w the expectation of the generalization error is zero: [gerr(w)]=0.

Let’s consider a solution that minimizes loss for a data set M:
w_m=argmin(L_m(w))=sum_n{a}/m.

Now, the expectation of the generalization error for that solution is not zero and equals
[gerr(w_m)]=-2[(a-[a])^2]/m. Note that it is negative and goes to zero when m goes to infinity.

Let’s consider a simple non-linear case.
The loss below is selected to simplify derivations and still good enough to make a statement.

Loss per sample is L(w)=(w^2-a^2)^2. Loss for m-point data set M is L_m(w)=sum_n{(w^2-a_n^2)^2}/m.

Minimum loss solutions are w_m^2=sum_n{a_n^2}/m. Also w_m=0 is a local maximum of the loss.
The generalization error at weight w is gerr(w)=2w^2([a^2]-sum_n{a_n^2}/m)+(sum_n{a_n^4}/m-[a^4]).
The expectation of the generalization error at a given weight is zero: [gerr(w)]=0.
However, the expectation of the generalization error for minimum loss solutions of the sampled data set is not zero:
[gerr(w_m)]=-2[(a^2-[a^2])^2]/m. It is negative and goes to zero for large m.
Let’s call this error -Err_m, so Err_m=2[(a^2-[a^2])^2]/m.

Now, let’s consider the effective loss by averaging the losses over Gaussian distribution with given (mu,s) - mean and std.
We use angular brackets for Gaussian averages: <L_m(w)> and <[L(w)]>.
Instead of dependency on weight these averages are functions of mu and s.

Then generalization error for a given mu, s is
gerr(mu,s)=<L_m(w)>-<[L(w)]>=-2mu^2(sum_n{a_n^2}/m-[a^2])-2s^2(sum_n{a_n^2}/m-[a^2])+(sum_n{a^4}/m-[a^4]).
Again, at given mu,s the expectation of generalization error [gerr(mu,s)] is zero.

Let’s consider solutions that minimize the effective loss for data set M w.r.t. mean and std:

d<L_m>/dmu=4mu(mu^2-(sum_n{a_n^2}/m-3s^2))=0,
 d<L_m>/ds=4s(3mu^2-(sum_n{a_n^2}/m-3s^2))=0.

There are trivial solutions that minimize the effective loss: s_m=0, mu_m^2=sum_n{a_n^2}/m.
And the critical point solution is: 3s_m^2=sum_n{a_n^2}/m, mu_m=0.

For trivial solutions the expectation of the generalization error [gerr(mu_m,s_m=0)]=-Err_m  - same as found above.
For the critical point solution the expectation of the generalization error [gerr(mu_m=0,s_m)]=-Err_m/3.

Now we can see that for the critical point the absolute value of the expectation of the generalization error is three times smaller than the error for trivial solutions.

While this is not a proof, it is a good example that supports the claim of the paper. The proof will be provided when it is finalized.

---

### Meta-Review · Area_Chair1 · 2018-11-25
**A good problem, but not well executed and communicated**

**Confidence:** 5
**Recommendation:** Reject

**Metareview:**

This paper studies a variational formulation of the loss minimization to study the solution that generalizes the most. An expectation of the loss wrt a Gaussian distribution is minimized to find the mean and variance of the Gaussian distribution. As the variance goes to zero, we recover the original loss, but for a higher value of variance, the loss may be convex. This is used to study the generalizability of the landscape.

Both objective and solutions of the paper are unclear and not communicated well. There is not enough citation to previous work (e.g., Gaussian homotopy exactly considers this problem, and there are papers that study the convexity of the expectation of the loss function). There are no experimental results either to confirm the theoretical finding.

All the reviewers struggle to understand both the problem and solutions discussed in this paper. I believe that the paper could become useful if reviewers' feedback is taken seriously to improve the paper.